# Bayesian inverse modeling and source location of an unintended I-131 release in Europe in the fall of 2011

Ondřej Tichý[1], Václav Šmídl[1], Radek Hofman[1], Kateřina Šindelářová[1], Miroslav Hýža[2], and Andreas Stohl[3]

[1]Institute of Information Theory and Automation, Czech Academy of Sciences, Prague, Czech Republic
[2]National Radiation Protection Institute, Prague, Czech Republic
[3]NILU: Norwegian Institute for Air Research, Kjeller, Norway

*Correspondence to:* Ondřej Tichý (otichy@utia.cas.cz)

**Abstract.** In the fall of 2011, iodine-131 (I-131) was detected at several radionuclide monitoring stations in Central Europe. After investigation, the International Atomic Energy Agency (IAEA) was informed by Hungarian authorities that I-131 was released from the Institute of Isotopes Ltd in Budapest, Hungary. It was reported that a total activity of 342 GBq of I-131 was emitted between September 8 and November 16, 2011. In this study, we use the ambient concentration measurements of I-131 to determine the location of the release as well as its magnitude and temporal variation. As the location of the release and an estimate of the source strength became eventually known, this accident represents a realistic test case for inversion models. For our source reconstruction, we use no prior knowledge. Instead, we estimate the source location and emission variation using only the available I-131 measurements. Subsequently, we use the partial information about the source term available from the Hungarian authorities for validation of our results. For the source determination, we first perform backward runs of atmospheric transport models and obtain source-receptor-sensitivity (SRS) matrices for each grid cell of our study domain. We use two dispersion models, Flexpart and Hysplit, driven with meteorological analysis data from the global forecast system (GFS) and from European center for medium-range weather forecasts (ECMWF) weather forecast models. Second, we use a recently developed inverse method, least-squares with adaptive prior covariance (LS-APC), to determine the I-131 emissions and their temporal variation from the measurements and computed SRS matrices. For each grid cell of our simulation domain, we evaluate the probability that the release was generated in that cell using Bayesian model selection. The model selection procedure also provides information about the most suitable dispersion model for the source term reconstruction. Third, we select the most probable location of the release with its associated source term and perform a forward model simulation to study the consequences of the iodine release. Results of these procedures are compared with the known release location and reported information about its time variation. We find that our algorithm could successfully locate the actual release site. The estimated release period is also in agreement with the values reported by IAEA and the reported total released activity of 342 GBq is within 99% confidence interval of the posterior distribution of our most likely model.

# 1 Introduction

In the fall of 2011, I-131 was detected in the atmosphere by the European Trace Survey Stations Network for Monitoring Airborne Radioactivity (Ring of 5, Ro5). The measured values were very low, up to a few tens of $\mu Bqm^{-3}$, close to the minimum detectable activity of the instruments. After the first findings in Austria and their subsequent confirmation by Czech laboratories, it was clear that these detections could not be explained by local sources. Hence, the International Atomic Energy Agency (IAEA) was informed on November 11 and launched an investigation. Detectable concentrations of I-131 were afterwards also measured by other laboratories, mainly in Central Europe (Atomic Energy Agency, 2011a). Based on the information provided by other Ro5 laboratories and a rough assessment of meteorological conditions, it was estimated that the source was likely located east of Austria and the Czech Republic. This was later confirmed when the IAEA Incident and Emergency Centre (IEC) was informed by the Hungarian Atomic Energy Authority (HAEA) (Atomic Energy Agency, 2011b) that I-131 had been released from the Institute of Isotopes (IoI) Ltd, Budapest, a facility that produces I-131 mainly for healthcare such as thyroid diagnosis. It is thought that a failure in the dry distillation process caused the emissions (Gitzinger et al., 2012). It was later reported that between September 8 and November 16, 2011, a total activity of 342 GBq of I-131 had been released from the institute, with a maximum release intensity between October 12 and October 14 of 108 GBq (Atomic Energy Agency, 2011b). The release is thought to have occurred through the 80 meters high stack of the institute. Since the released activity was below the institute's authorized annual radioactive release limit and I-131 concentrations in the air were very low, IAEA stated that the situation did not pose a health risk.

Although some ambient concentration measurements are available for this case, they are quite sparse, poorly resolved in time (typically sums over 7 days), and cover many orders of magnitude. This makes an analysis of the impact of the event based on measurement data alone very difficult. For example, if no measurements are available in the area of the largest impact, the severity of the event may be grossly underestimated. Given accurate release information, atmospheric transport models can simulate the radioiodine dispersion and give a more comprehensive view of the situation than the measurements alone. For instance, simulations with atmospheric transport models were used previously to study the distribution of radioactive material after the Chernobyl, e.g. Brandt et al. (2002); Davoine and Bocquet (2007), and Fukushima Dai-ichi nuclear accidents, e.g. Morino et al. (2011); Stohl et al. (2012); Saunier et al. (2013). Simulations were also already made for the I-131 release from IoI in 2011 (Leelőssy et al., 2017). However, the agreement between the results of simulations and real measurements needs to be carefully evaluated since simulations often suffer from inaccuracies in meteorological input data or model parametrizations. The largest errors in such simulations are arguably caused by uncertainties in the source term of the release, i.e., the rate of emissions into the atmosphere as a function of time. However, the release term is often not known and its determination can be particularly difficult in case of a nuclear accident since the release can last for a long time and its intensity can vary by orders of magnitude.

To our best knowledge, the exact source term in the case of the Hungary iodine release in 2011 is unknown and only approximate and vague information is available (Gitzinger et al., 2012). For lack of information on the operating conditions of the isotope production facility, we cannot use the so-called bottom-up approach where the source term is quantified based

on understanding and modeling of the emission process. Therefore, in this paper we use the so-called top-down approach (Nisbet and Weiss, 2010), which combines ambient concentration measurements with an atmospheric transport model and an optimization algorithm to determine the source term. This approach is also called inverse modeling. The source term is typically estimated as a result of optimization of the difference between the measurements and corresponding simulated sensor readings

predicted by the atmospheric transport model. Due to insufficient information provided by the measurement data, the problem has to be regularized using a penalty function (Seibert, 2000; Eckhardt et al., 2008), the maximum entropy principle (Bocquet, 2005), or a variational Bayesian approach (Tichý et al., 2016). All these methods assume that the measurement vector can be described as a linear model with a source-receptor-sensitivity (SRS) matrix (calculated using an atmospheric dispersion model, e.g. Seibert and Frank (2004)) and unknown source term vector.

The range of possible regularization techniques starts with positivity constraint of the source, simple Tikhonov penalty, see e.g. Davoine and Bocquet (2007), and additional enforcement of temporal and/or spatial smoothness of the release, see e.g. (Eckhardt et al., 2008). Interpretation of the regularization as a prior covariance matrix allows its estimation. Different methods exist for parametrizations of both the measurements covariance matrix and source term covariance matrix. Winiarek et al. (2012) parameterize each covariance matrix using one common parameter on its diagonal. A similar model was also studied

by Michalak et al. (2005) with different diagonal entries and by Berchet et al. (2013) with full unknown covariance matrices, however, with convergence issues since too many parameters need to be estimated in this case. Therefore, non-diagonal matrix elements are often parametrized using autocorrelation parameters that link covariance in space and/or time (Ganesan et al., 2014; Henne et al., 2016). In this paper, we follow a previously developed approach (Tichý et al., 2016) where the source term covariance matrix is adaptively estimated within the estimating procedure using a variational Bayes methodology (Šmídl and

Quinn, 2006) or Gibbs sampling (Ulrych and Šmídl, 2017).

An application of the inverse modeling problem is the source location problem. If the release site is unknown, the inverse modeling is performed for many potential release sites and their likelihood of being the correct site is compared. The simplest scenarios assume a constant release rate (Annunzio et al., 2012; Zheng and Chen, 2010; Ristic et al., 2016) or even steady wind field (Liping et al., 2013). However, these are not very realistic assumptions, especially not for complex emission scenarios

with continental-scale impacts.

Typically, the inverse modeling problem is recast as an optimization problem such as the weighted linear or nonlinear least squares, e.g. Singh and Rani (2014); Matthes et al. (2005), simulated annealing, e.g. Thomson et al. (2007), or pattern search method, e.g. Zheng and Chen (2010). Cervone and Franzese (2010) studied several error functions to identify suitable measures and cost functions for optimizations and Kovalets et al. (2011) used a fluid dynamics model to build up a cost function

which could be optimized. These methods can be inconvenient due to problematic convergence and limited information on the uncertainty of the results. Often, they provide only point estimates. Full posterior probability densities are provided using Bayesian techniques where the prior model is typically constructed as an alternative to the cost function in the optimization approach. Very popular Bayesian inference techniques are random search algorithms such as Markov chain Monte Carlo (MCMC) methods. Examples for this type of approach are Keats et al. (2007) and Senocak et al. (2008) where also wind field

parameters are estimated along with the source term parameters, or Delle Monache et al. (2008) who studied the Algeciras

accidental release with the assumptions that the source geometry and release time are known. Another Bayesian formulation and inference using maximum entropy principle was proposed by Bocquet (2007) where the source term is modeled as three-dimensional (area plus time); hence, the source term integrated over time and area is obtained. This approach was tested for both cases of the European tracer experiment (ETEX) (Krysta et al., 2008) and compared with the maximum posterior estimator by Bocquet (2008) with further non-Gaussian assumptions such as positivity or boundedness. Recently, a likelihood-free approximate Bayesian computation method for the localization of a biochemical source was proposed by Ristic et al. (2015) where multiple dispersion models can be used and even weighted using Bayesian model selection. An extensive review of source term estimation and location is available in (Hutchinson et al., 2017).

Recently, a Bayesian inverse method called least-squares method with adaptive prior covariance (LS-APC) was proposed (Tichý et al., 2016) using Variational Bayes (VB) approximation. The method was validated on the basis of the ETEX experiment and it was shown that the dependency on manual selection of model parameters is lower than in case of its predecessors. The key advantage of the VB approach is its fast evaluation which makes it suitable for calculation of many possible source locations. However, the method is known to underestimate uncertainty, therefore, we will also use a more accurate approximation of the posterior distribution based on the Gibbs sampler (GS) (Ulrych and Šmídl, 2017).

In this paper, we use the LS-APC method for inversion for the case of the iodine release in Hungary in 2011. Moreover, we derive the Variational Bayesian model selection for the LS-APC model. Using this methodology, we can compare the reliability of each SRS matrix from the selected spatial domain at reasonable computational cost. The same methodology can be used to quantify uncertainty in the evaluation of the SRS matrix. Specifically, if several possible variants of the SRS matrix computation are available, the Bayesian model selection can evaluate their posterior probability, providing an objective guideline for selection of the most likely dispersion model or weather data. In this study, we evaluate probability of the SRS matrices obtained using backward runs of the dispersion models Flexpart (Stohl et al., 2005) and Hysplit (Draxler and Hess, 1997) which were based on meteorological input data from GFS meteorological fields with resolution of $0.5° × 0.5°$ in the case of Flexpart and from GFS meteorological fields with resolutions $0.5° × 0.5°$ and $1° × 1°$ and ECMWF meteorological fields with resolution of $0.5° × 0.5°$ in the case of Hysplit. We identify the most probable release location and derive the corresponding estimated source term. With a low number of selected locations, we run a more expensive approximation of the model based on Gibbs sampling which is more computationally demanding. Using this source information, we perform a forward run and produce a I-131 dose map for Europe that can be used for impact assessment.

## 2   Measurement data

Iodine can exist in the atmosphere both as a gas and in the aerosol phase. Measurements of particulate phase I-131 were made at several stations of the Ro5 network, which is an informal information group established in 1983 for the purpose of rapidly exchanging data on occasional enhanced concentrations of man-made radionuclides at trace levels. In total, 117 I-131 measurements from 11 different sampling sites in Central Europe (see details in Table 1) obtained from September to November

2011 were used in this study. As an example, measurements for the whole period from the Budapest and Praha stations are displayed in Fig. 1.

Atmospheric aerosol sampling was performed using various types of high volume samplers with flow rates ranging from 150 to 900 $\text{m}^3$/h. In these devices, the air is filtered through glass-fiber or polypropylene filters which capture the radioactive aerosol with a high efficiency. As the laboratories operate under their own monitoring plans, sampling intervals differ both in length and starting day. In general, filters are changed every 3–7 days under normal conditions. Only in case of an emergency situation, the sampling period would be shortened.

After the sampling completion and decay of short-lived radon decay products, the filters are measured without additional chemical preparation in laboratories equipped with a high resolution gamma ray spectrometer. Since I-131 emits 364 keV photons with an intensity of 81%, it allows a reasonably sensitive determination by a high-purity germanium (HPGe) spectrometer. In such a measurement arrangement, it is possible to achieve detection limits of several µBq/$\text{m}^3$ but at the cost of a rather poor time resolution. Considering the 8.02 day half-life of I-131, the resulting activity value has to be decay corrected which requires the assumption that the concentration in the air was constant during sampling.

## 3 Inverse modeling

We follow the concept of linear modeling of the atmospheric dispersion using a source-receptor sensitivity (SRS) matrix, see e.g. Seibert (2001); Wotawa et al. (2003); Seibert and Frank (2004). In this approach, an atmospheric transport model is used to provide the linear relationship between sources and atmospheric concentrations. By assuming a release $x_i$ from the release site at time $i$, we can calculate the concentration response at a receptor $y_j$ at time $j$. Notice that the simulated concentration response can be compared directly with measured concentrations at the receptor. The ratio $m_{ij} = y_j/x_i$ defines the source receptor sensitivity. Collecting all possible release times in vector $\mathbf{x} \in \mathbf{R}^n$ and all possible receptor responses at all measurement sites and times into vector $\mathbf{y} \in \mathbf{R}^p$ we obtain a linear model

$$\mathbf{y} = M\mathbf{x} + \boldsymbol{\epsilon}, \tag{1}$$

where $M \in \mathbf{R}^{p \times n}$ is a SRS matrix and $\boldsymbol{\epsilon} \in \mathbf{R}^p$ is an observation error including both model and measurement errors, where the model error contained in matrix $M$ is projected onto the observation vector. This concept of SRS is quite universal and can be applied with both Lagrangian and Eulerian transport models in both forward and backward runs (Seibert and Frank, 2004). However, the assumption of linearity is justified only for passive tracers and substances which do not undergo nonlinear chemical transformations – which is largely the case for iodine, which is thought to have mainly linear removal processes (radioactive decay and wet and dry deposition to the surface).

An estimate of the unknown vector $\mathbf{x}$ can be obtained using minimization of the model error (1). However, a Bayesian approach provides more informative results since it evaluates the full posterior density of the unknown. The high computational cost of conventional Monte Carlo evaluation methods can be avoided by using an approximation technique known as Variational Bayes. This has been analyzed in detail by Tichý et al. (2016), where a computationally efficient algorithm was presented. One

of the key advantages is that all parameters of the regularization are estimated together with the source term. In this paper, we provide an approximate formula for the evaluation of the marginal likelihood of the model, which is essential for Bayesian model comparison (Bernardo and Smith, 2009). In effect, this technique allows to compare the likelihood of different matrices $M$ which could describe atmospheric dispersion from different possible source locations or could originate from different atmospheric dispersion models.

Before reviewing the full probabilistic model, we would like to illustrate its relation to the conventional cost optimization. Consider the quadratic norm of the residues of (1)

$$J = \omega_0^{-2} \left(M\mathbf{x} - \mathbf{y}\right)^T \left(M\mathbf{x} - \mathbf{y}\right), \tag{2}$$

with selected parameter $\omega_0$. The estimate $\langle \boldsymbol{x} \rangle$ can be obtained by minimizing the cost $J$ (Eq. 2) plus additional regularization terms. In probabilistic interpretation, minimization of Eq. 2 is equivalent to maximization of the likelihood function

$$p(\mathbf{y}|\mathbf{x}) = \mathcal{N}\left(M\mathbf{x}, \omega_0^{-1} I_p\right) \propto \exp\left(-\frac{1}{2}\omega_0 \left(M\mathbf{x} - \mathbf{y}\right)^T \left(M\mathbf{x} - \mathbf{y}\right)\right), \tag{3}$$

where $\mathcal{N}(\boldsymbol{\mu}, \Sigma)$ denotes a multivariate Gaussian distribution with mean $\boldsymbol{\mu}$ and covariance matrix $\Sigma$, $I_p$ is the $p \times p$ identity matrix, and symbol $\propto$ denotes equality up to the normalizing constant. In this case, $\Sigma = \omega_0^{-1} I_p$ and $\omega_0$ is known as the precision parameter. The normalization constant is irrelevant for maximization. However, it will become important for estimating the precision parameter $\omega_0$. Due to the requirement of normalization, the Bayesian method allows to estimate parameters of the prior distributions (which define the regularization terms in the cost formulation). To distinguish between selected and estimated parameters, we denote all preselected parameters with subscript 0 and estimated model parameters without the subscript.

After reviewing the selected Bayesian inverse method, we will derive a lower bound on its marginal likelihood which will be used for selection of the most suitable model structure. Specifically, we will use this tool to select from multiple SRS matrices arising from different settings of the dispersion model. Multiple SRS matrices may arise, e.g., when multiple atmospheric transport models are available, when varying model parameters, when multiple meteorological input data are available, or when SRS matrices are computed for each potential release site. The marginal likelihood measure is able to select the most suitable model, with natural penalization for complex models due to the principle of marginalization. Thus, the influence of the estimated tuning parameters (hyper-parameters of the prior) is minimized.

## 3.1 Review of model LS-APC

The probabilistic model of Tichý et al. (2016) is briefly reviewed in this Section. The likelihood function is considered to be Gaussian (3) with standard deviation $\omega$ being considered as unknown. Thus, we need to select its prior distribution. We select the gamma distribution due to its conjugacy with Gaussian likelihood (Tipping and Bishop, 1999):

$$p(\mathbf{y}|\mathbf{x}, \omega) = \mathcal{N}\left(M\mathbf{x}, \omega^{-1} I_p\right), \tag{4}$$

$$p(\omega) = \mathcal{G}\left(\vartheta_0, \rho_0\right), \tag{5}$$

where $\vartheta_0, \rho_0$ are chosen constants. These constants are needed for numerical stability, however, they are set as low as possible such as to $10^{-10}$ to provide a non-informative prior.

The prior distribution of the source term $\mathbf{x}$ is designed to encourage three properties: (i) non-negativeness of all elements of $\mathbf{x}$, (ii) sparsity, i.e., the element is zero unless there is sufficient information on the opposite, and (iii) smoothness, i.e. that rapid changes in the temporal profile are possible but not frequent. These properties are encoded into a hierarchical prior model

$$p(x_{j+1}|x_j, l_j, v_j) = t\mathcal{N}\left(-l_j x_j, v_{j+1}^{-1}, [0, \infty]\right), \quad \text{for } j = 1, \ldots, n-1, \tag{6}$$

$$p(v_j) = \mathcal{G}\left(\alpha_0, \beta_0\right), \quad \text{for } j = 1, \ldots, n, \tag{7}$$

$$p(l_j|\psi_j) = \mathcal{N}\left(-1, \psi_j^{-1}\right), \quad \text{for } j = 1, \ldots, n-1, \tag{8}$$

$$p(\psi_j) = \mathcal{G}\left(\zeta_0, \eta_0\right), \quad \text{for } j = 1, \ldots, n-1, \tag{9}$$

where $t\mathcal{N}(\mu, \sigma, [a, b])$ denotes the truncated Gaussian distribution on support $[a, b]$, $l_j$ is a parameter modeling the smoothness, i.e. the relation between neighboring elements of the source term, and $v_j$ is its precision parameter. The prior for element $x_1$ is $p(x_1|v_1) = t\mathcal{N}\left(0, v_1^{-1}, [0, \infty]\right)$. The prior has constants $\alpha_0, \beta_0, \zeta_0, \eta_0$ that need to be selected. Good performance of the prior was reported with a non-informative choice of $\alpha_0, \beta_0$, e.g., $10^{-10}$. The prior constants $\zeta_0$ and $\eta_0$ are selected as $10^{-2}$ to favor a smooth solution, see discussion in (Tichý et al., 2016).

## 3.2 Model uncertainty

The original LS-APC model (4)–(9) assumes uncertainty only in the source term $\mathbf{x}$ and its hyper-parameters. However, in real scenarios, the uncertainty is also present in the SRS matrix due to inaccurate meteorological data and/or inaccurate parameters of the dispersion model. Exact modeling of these uncertainties is too complex, therefore, we use an approximation using discrete variable. Specifically, we assume that we have a finite set of SRS matrices, $\mathcal{M} = \{M_1, M_2, \ldots, M_r\}$ obtained by different versions of the dispersion models and/or different meteorological data. Uncertainty in the SRS matrix and the potential bias of the results is thus reduced by estimating the probability that the data were generated by each of the tested SRS matrix. The result is thus a principled way how to select the most likely dispersion model and meteorology for a particular data set.

## 3.3 LS-APC model inference

The LS-APC model is a hierarchical Bayesian model designed to estimate its hyper-parameters from the data. For a given model (SRS matrix) $M$, the task of the inference is to use the Bayes rule to find the posterior distribution

$$p(\mathbf{x}|\mathbf{y}, M) = \frac{p(\mathbf{y}, \mathbf{x}|M)}{p(\mathbf{y}|M)}, \tag{10}$$

in which all nuisance parameters (i.e. $\omega, v, l, \psi$) have been marginalized (integrated out). The denominator of (10) is known as marginal likelihood and it is essential in evaluation of the probability of the model represented by the SRS matrix from the set $\mathcal{M} = \{M_1, \ldots, M_r\}$. The probability that the observed data were generated from the $k$th model, $M_k$, $k = 1, \ldots, r$ can be formally obtained from the Bayes rule

$$p(M = M_k|\mathbf{y}) \propto p(M = M_k)p(\mathbf{y}|M_k). \tag{11}$$

Here, symbol $\propto$ denotes equality up to a multiplicative constant, and $p(M = M_k)$ denotes prior probability of the $k$the model. In our case we assume that all models have equal prior probability. Evaluation of (10) and (11) is intractable and will be approximated by the Variational Bayes and Gibbs sampling methods.

### 3.3.1 Variational Bayes inference

Under the VB approximation (Šmídl and Quinn, 2006), the posterior distributions are found in the same form as their priors (6)–(9) and their moments are determined by an iterative algorithm which is available in Matlab code as a supplement of Tichý et al. (2016). However, the value of the marginal likelihood $p(\mathbf{y}|M)$ is not available analytically and no approximation was presented in (Tichý et al., 2016). The method will be referred here as the LS-APC-VB algorithm.

Approximation of the marginal likelihood (11) using Variational Bayes methodology is computed as

$$p(M = M_k|\mathbf{y}) \propto p(M = M_k)\exp\left(\mathcal{L}_{M_k}\right),\ k = 1,\ldots,r, \tag{12}$$

where $\mathcal{L}_{M_k}$ is a variational lower bound on $p(\mathbf{y}|M_k)$ (Bishop, 2006) given as

$$\mathcal{L}_{M_k} = \int p(\mathbf{x},\Upsilon,L,\boldsymbol{\psi},\omega|M_k)\,p(M_k)\ln\frac{p(\mathbf{y},\mathbf{x},\Upsilon,L,\boldsymbol{\psi},\omega,M_k)}{p(\mathbf{x},\Upsilon,L,\boldsymbol{\psi},\omega|M_k)p(M_k)}\mathrm{d}\mathbf{x}\mathrm{d}\Upsilon\mathrm{d}L\mathrm{d}\boldsymbol{\psi}\mathrm{d}\omega, \tag{13}$$

where $\mathbf{x},\Upsilon,L,\boldsymbol{\psi} = [\psi_1,\ldots,\psi_{n-1}],\omega$ are variables of the LS-APC model driven with the SRS matrix $M_k$ (variables $\Upsilon$ and $L$ are matrices defined in the supplementary material). Eq. (13) can be seen as a term composed of expected values (denoted as $\mathrm{E}[]$ with respect to distribution of the variable in its argument) so that

$$\mathcal{L}_{M_k} = \mathrm{E}\left[\ln p(\mathbf{y},\mathbf{x},\Upsilon,L,\boldsymbol{\psi},\omega,M_k)\right] - \mathrm{E}\left[\ln\tilde{p}(\omega)\right] - \mathrm{E}\left[\ln\tilde{p}(\mathbf{x})\right] - \mathrm{E}\left[\ln\tilde{p}(\Upsilon)\right] - \mathrm{E}\left[\ln\tilde{p}(L)\right] - \mathrm{E}\left[\ln\tilde{p}(\boldsymbol{\psi})\right], \tag{14}$$

where $p(\mathbf{y},\mathbf{x},\Upsilon,L,\boldsymbol{\psi},\omega,M_k)$ is the joint distribution of likelihood (4) and prior probability distributions (6)–(9), and $\tilde{p}()$ are posterior probability distributions. These terms are given in the supplementary material.

### 3.3.2 Gibbs sampling inference

An alternative approximation of the posterior (10) is obtained using Gibbs sampling (GS). The method is closely related to the VB method (Ormerod and Wand, 2010) using the same forms of posterior with different interpretation. While the Variational Bayes approximation is looking for a good fit of parametric form, the Gibbs sampling generates samples from the conditional distribution and approximates the posterior by an empirical distribution on these samples. It has been applied to the LS-APC model by Ulrych and Šmídl (2017). In practical terms, the GS yields a more accurate approximation, however, at the cost of a much higher computational burden. While the VB method converges in less than 100 iterations, the GS method needs about 1000000 samples to obtain reliable estimate (one sample takes roughly the same CPU time as one iteration of VB). However, the main advantage is that the GS method converges to the true posterior, while the VB method may converge to a local approximation. The method will be referred here as the LS-APC-GS algorithm.

## 4 Atmospheric transport modeling

The SRS matrices in this work were computed using backward runs of two alternative models, namely Hysplit (Draxler and Hess, 1997) and Flexpart (Stohl et al., 2005). As the domain of interest we chose the region spanning from 5° E to 30° E in longitude and from 40° N to 65° N in latitude covering most of Europe and parts of the Mediterranean Sea. Horizontally, the domain was discretized into 2500 grid cells with resolution $0.5° \times 0.5°$ which approximately corresponds to $45\,\text{km} \times 55\,\text{km}$ at the latitude of Budapest. Vertically, there is no discretization of the domain and sensitivities are calculated for a layer 0-300 m above ground which allows for both ground and somewhat elevated releases (e.g., through the stack of the isotope production facility). Mixing heights are often higher than 300 m, in which case the result is not very sensitive to the choice of the depth of this layer. Temporal resolution of the source was set to 1 day and we assume that the release occurred during a 91 day time window starting on 1 September 2011.

As a result, the domain was discretized into 227500 spatial-temporal sources for which their possible contributions to all samples must be calculated. Since the number of candidate sources is much higher than the number of measurement samples, the SRS matrices were obtained using backward runs of the model from the sampling sites. One backward run was started exactly at the point location of each measurement site and for each period corresponding exactly to a measurement sample. Each of the 117 backward runs corresponding to the 117 available measurements provided a SRS matrix of a particular sample to all candidate spatio-temporal sources in our domain. Since we a-priori assume that the release occurred from a point source (i.e., a single horizontal grid cell), we can calculate SRS fields from a single grid cell at once which allows parallelization of the computations. We end up with 2500 SRS matrices (one for each of the $50 \times 50$ model grid cells) of dimension $117 \times 91$ from each transport model.

Radioiodine can be present in the atmosphere as molecular $I_2$, as organic iodide, or as iodide salts. The former two are expected to exist as gases, while the latter is an aerosol. In which form iodine is released to the environment from a nuclear facility depends on its operating conditions (Simondi-Teisseire et al., 2013). Iodine chemistry in the atmosphere is complex and can involve, for instance, chemical transformation of the different compounds and particle formation (Saiz-Lopez et al., 2012). As every compound has its own scavenging efficiency, both with respect to dry and wet deposition, accurate modeling of iodine is complicated. We chose a simple approach for our modeling, namely assuming that all released I-131 was in particulate form, which most probably dominated the release. This is also justified by the fact that all of the measurements we have available were made for particulate iodine only. Consequently, in both models, I-131 was simulated as an aerosol. In Flexpart, parameters of the dry and wet deposition were set to default values for I-131 in the Flexpart 9.2 species library and radioactive decay (ingrowth during backward runs) was calculated on the fly. In Hysplit, parameters of the dry and wet deposition were set to default values for aerosol I-131, except for predefined dry deposition velocity which was set to 5.7 $\text{mms}^{-1}$ according to measurements of Takeyasu and Sumiya (2014). Hysplit calculated with I-131 radioactive decay half-life of 8 days. Our inverse modeling would thus not capture gaseous I-131, which may have been co-emitted, except indirectly if some of this gaseous I-131 condensed on or formed particles that were subsequently measured. Our results are thus lower estimates of the total I-131 release, but the bias is probably not very large.

## 4.1 Flexpart

Flexpart (FLEXible PARTicle dispersion model) is a scientific model used worldwide by many research groups and also operationally, e.g. at CTBTO for routine atmospheric backtracking (Kalinowski et al., 2008). In this work we used version 9.2 (Stohl et al., 2005). Runs were forced with GFS meteorological fields with $0.5° \times 0.5°$ horizontal resolution and 26 vertical layers and temporal resolution of 3 hours. During all calculations, the convection scheme was enabled in Flexpart for more realistic simulation of vertical air mass fluxes when convective conditions are encountered (Forster et al., 2007).

Simulations in Flexpart can be carried out on two different output grids in a single run. The so called mother grid is usually a global grid with coarser resolution whereas the nested grid is a smaller subdomain with higher horizontal resolution (vertical resolution must be the same for both grids). Our domain of interest was a nested output grid with horizontal resolution $0.5° \times 0.5°$ whereas the global grid with resolution $1° \times 1°$ was the mother grid. The simulations accounted for dry deposition using a resistance method. Wet scavenging was accounted for with a scheme that distinguishes between in-cloud and below-cloud scavenging.

## 4.2 Hysplit

The Hysplit (HYbrid Single-Particle Lagrangian Integrated Trajectory) model is a model widely used to simulate atmospheric transport and dispersion on various levels of complexity. Its applications range from simple estimation of forward and backward trajectories of air parcels, to advanced modeling of transport, dispersion and deposition of air masses on large domains. Hysplit adopts a hybrid approach combining the Lagrangian (moving frame of reference for diffusion and advection) and Eulerian (fixed model grid for calculation of air concentration) model methodologies. In this study we applied Hysplit model version 4 (Draxler and Hess, 1997, 1998; Draxler and Rolph, 2003; Stein et al., 2015).

The model was forced with GFS analyses with horizontal resolution of $0.5° \times 0.5°$, 26 vertical layers and 6-hourly temporal resolution. The model domain covered most of the European continent. The Hysplit model was also forced with GFS analyses with horizontal resolution of $1° \times 1°$, 26 vertical layers and 6-hourly temporal resolution, to test the sensitivity of the source re-construction to meteorological input data resolution. This data set was only available in a format suitable for Hysplit but not for Flexpart. The resolution of the output grid was the same as used with Flexpart, i.e., $0.5° \times 0.5°$. The Hysplit model was also forced with the ERA-Interim reanalysis (Dee et al., 2011) data from the European Center for Medium-range Weather Forecast (ECMWF) with $0.5° \times 0.5°$ horizontal resolution, 36 vertical layers and temporal resolution of 6 hours.

## 5 Results and discussion

In this Section, we apply the Bayesian inverse modeling method introduced in Sect. 3 to iodine measurements described in Sect. 2 and computed SRS matrices from Sect. 4 for all four cases: (i) Flexpart driven with the GFS analyses with the resolution $0.5° \times 0.5°$ (Flexpart-GFS-0.5), (ii) Hysplit driven with the GFS analyses with the resolution $0.5° \times 0.5°$ (Hysplit-GFS-0.5), (iii) Hysplit driven with the GFS analyses with the resolution $1° \times 1°$ (Hysplit-GFS-1.0), and (iv) Hysplit driven with the ECMWF

analyses with resolution $0.5° \times 0.5°$ (Hysplit-ECMWF-0.5). First, we will study the problem of source location, and after that we will discuss the source term as a function of time for the most probable source location.

## 5.1 Source location

The LS-APC-VB inversion method, described in Sect. 3, was applied to each grid cell in our domain (notice that each grid cell is a candidate source location) for each combination of atmospheric transport model and meteorological input data. Hence, our set of SRS matrices is defined as $\mathcal{M} = \{M_{(i,j,m)}; i = 1, \ldots, 50, j = 1, 50, m = 1, \ldots, 4\}$ where $i, j$ are coordinates of the $(i, j)$th tile on the map and $m$ is the number of specific combination of atmospheric transport model driven with meteorological input data. For each SRS matrix from the set $\mathcal{M}$, the method also provides the variational lower bound $\mathcal{L}_{M_{(i,j,m)}}$, Eq. (14), which correspond to the probability that the release happened in grid cell $(i, j)$ for the given atmospheric model. Note that no prior information on source location, $p\left(M_{(i,j,m)} = M\right)$ in Eq. (12), is used which is equal to omitting of this term due to proportional equality in the equation. The results are presented in Fig. 2 for Flexpart-GFS-0.5 (top left), Hysplit-GFS-0.5 (top right), Hysplit-GFS-1.0 (bottom left), and Hysplit-ECMWF-0.5 (bottom right).

In all four cases, the source location mechanism of the LS-APC-VB method works very well and the maxima of the variational lower bound $\mathcal{L}_{M_{(i,j,m)}}$ are close to the true location of the IoI. Note that the exact location of the IoI is 18.96° E and 47.49° N which is in the corner of a grid cell in the case of 0.5° resolution; hence, we assume all results close to this point to be very good. In the case of Flexpart-GFS-0.5, the estimated release site is on the edge and south-east of the actual release site. For both Hysplit-GFS cases with resolutions of 1.0 and 0.5, respectively, the release site is found on the edge and north-west of the actual release site, while when using Hysplit-ECMWF-0.5, the estimated release site is north-east and on the edge of the actual release site. In summary, the release site was well estimated using all atmospheric models in tandem with LS-APC-VB algorithm. In all four cases, some uncertainty remains especially to the south of the IoI where no measured data are available while in the north, the uncertainty is very small because the relatively dense measurement network there effectively excludes the possibility of a source in this region. This is a typical problem of inverse methods when the geometry of the sampling network is sub-optimal and the source location is not surrounded by stations. This situation is similar to tomographic reconstructions, e.g., in medical applications, where the reconstruction quality is always best when measurements can be taken all around the phantom. Nonetheless, we conclude that the LS-APC-VB method provides reasonable source locations in all studied cases, even with the sub-optimal distribution of measurement stations.

We would like to point out that the Bayesian model selection allows to compare the likelihood of models for any set of matrices $M_{(i,j,m)}$, even if they are from different dispersion models and meteorological input data. The global maximum of the model likelihood for all cases is achieved with the Hysplit-GFS-0.5 configuration, see colorbars in Fig. 2.

The Gibbs sampling is computationally too expensive to run it for the full set of potential source locations. However, we ran it for a very small neighborhood around the best location identified with the LS-APC-VB method. The results closely correspond to those of the VB approximation, with occasional changes between the best and second best location. The differences in log-likelihood between models are smaller than in the case of the VB method. The main difference from VB is that the GS approach selects the most likely release to be that of the best location for the Hysplit-GFS-1.0 model.

## 5.2 Source term estimation

With selected location of the release, we proceed to estimate the release profile using both approximations, the VB method and GS method. Source term estimates for the most likely locations obtained by the VB approximation for each dispersion model are given in Fig. 3. Full posterior densities are reported via their mean value (denoted by blue lines) and 95% highest posterior density regions (gray filled region). Notice that the computed total sums of activity with two sigma uncertainty bounds are also reported in Fig. 3. Hysplit-GFS-0.5 is the most likely of the four models according to the VB approximation. This can be understood from the scatter plots between measured data $\mathbf{y}$ and reconstructed signal $M\mathbf{x}$ in Fig. 4 for Flexpart-GFS-0.5 (top left), Hysplit-GFS-0.5 (top right), Hysplit-GFS-1.0 (bottom left), and Hysplit-ECMWF-0.5 (bottom right). Note that significantly lower marginal log-likelihoods of the Flexpart-GFS-0.5, Hysplit-GFS-1.0, and Hysplit-ECMWF-0.5 models reported in Fig. 2 and subsequent differences in source terms are due to only two measurements that are not explained well in the reconstruction. All other measurements are explained well.

The same data were processed using the LS-APC-GS method which provides results in the form of samples from the posterior distribution of the source term. The best values of the marginal likelihood for this approximation was obtained for model Hysplit-GFS-1.0. The posterior distributions of the source term for each of the tested models is displayed in Fig. 5 in the same layout as for the VB approximation. The result is a superposition of $10^6$ samples of the source terms. Due to low number of data, all scale parameters have posterior distributions with long tails resulting in a high number of samples with large release amounts which can be considered as outliers. The outliers have a strong impact on the mean value and, therefore, we will report the results in terms of the median (50th percentile) and uncertainty bounds in the form of 5th and 95th percentiles. Selection of a single source term, e.g. for computation of the scatter plot is problematic.

With respect to the time variation of the release, all source terms estimated by the VB method have an emission activity peak around the reported maximum activity period from October 12 to October 14, confirming this aspect of the official report. The main difference between the VB and GS approximations of the source term estimation is that the results of the VB approach are concentrated around a selected mode of the posterior distribution, while the GS approach considers all possible modes. Therefore, the GS results are a collection of many possible profiles. The posterior distribution in the period of October 12 to October 14 is not so narrow but contains a smooth bump. This is due to low informativeness of the data on temporal resolution, since the sampling period of the measurements is 7 days for the majority of the data. The estimates provided by the GS method also provide higher values of the total release amount than the VB method. We conjecture that this is due to the property of the VB approximation to yield a zero source term when the measurements are insensitive to its choice. The posterior distribution of the Gibbs sampler has also maximum at zero, but the median is positive. See Ulrych and Šmídl (2017) for discussion.

## 5.3 Discussion of the source terms

The officially reported total release activity was 342 GBq with a maximum release intensity between October 12 and October 14 of 108 GBq and a total release period from September 8 until November 16 (Atomic Energy Agency, 2011b). Compared to the official estimate, all VB and GS estimates (except for the VB solution for Hysplit-GFS-1.0) overestimate the total released

activity but are in the same order of magnitude as the official estimate. The reported value is around the first percentile of the Gibbs sampling results for the most likely model. Since these values are based on measurements of particulate iodine only, the estimates are lower bounds for the total release which may also have included radioiodine gas. Moreover, the results are subject to many unmodeled uncertainties which are now discussed.

First, one has to consider uncertainty due to the long sampling period of the measurements, mostly 7 days. This may lead to a large uncertainty in estimated source terms since the inversion method tries to capture a source term with resolution of one day from such a time-insensitive measurements. The second source of uncertainty is relatively coarse discretization of the studied domain and the proximity of the IoI facility and the measuring station Budapest (approximately 10 kilometers). Since concentration gradients cannot be resolved within one grid cell, the inversion may try to compensate this by overestimation

of the source term to fit the Budapest measurements. The third source of uncertainty of the source term are the selected atmospheric dispersion models (and their parameterizations). For example, both atmospheric transport models may simulate too short lifetime of particulate iodine. This, as for many other models, was found for Cs-137 attached to particles after the Fukushima Dai-ichi accident (Kristiansen et al., 2016). The inversion would probably try to compensate a too strong loss of mass by increasing the emitted amount. The fourth source of uncertainty are the input data from meteorological reanalysis.

As shown by Leelőssy et al. (2017), the example meteorological situation on November 4 in 2011 in Central Europe was very complex with low-level inversion where the winds below the inversion level were significantly different than the winds above. Subsequently, if boundary layer heights were systematically too high, simulated ground-level concentrations may be systematically too low. This would be probably compensated by the inversion with a too large emitted amount. In these and other complex situations, different models may provide very different performances (Leelőssy et al., 2017).

Given all these uncertainties and the fact that also in our study, different atmospheric transport models driven with different meteorological reanalyses provide also different source terms, one should be cautious in comparing the total estimated release with the reported release amount. An agreement of the total amount of released I-131 within one order of magnitude may be the maximum which can be expected. This is reflected by the large uncertainty ranges obtained with our method. A positive result is that the models selected by the marginal likelihood provide results closer to the reported values than the other models.

Our model ensemble is too small to fully capture the uncertainty related to the choice of the dispersion model or meteorological input data. Nevertheless, our small ensemble shows that the results are quite sensitive to the choice of the model. Particularly noteworthy is the large difference between Hysplit-GFS-1.0 and Hysplit-GFS-0.5, since these use the same dispersion model and meteorological input data, except for the resolution of the latter.

This high sensitivity is at least partly related to the small number of available I-131 measurements. The inversion may take

advantage of certain model features to fit the model results to the few measurements. Such "overfitting" by exploiting particular model characteristics is less likely to be successful for a larger measurement data set. Clearly, more measurements are needed for a more reliable source term estimation. Nevertheless, the estimated source terms are of the right order of magnitude and the estimated release periods between early September and mid-November correspond well with the reported probable release period of September 8 to November 16 (Atomic Energy Agency, 2011b).

## 5.4    Forward modeling of the iodine release

Using the estimated source location and source term, we can perform a forward run of the model and study the simulated consequences of the accidental release. For this purpose, we identify the most probable location of the release from all cases evaluated by the LS-APC-VB method (Fig. 2), which is the location with center at 18.75° E and 47.75° N obtained with the Hysplit-GFS-0.5 configuration with log-likelihood up to 340. Therefore, we perform a forward run with the Hysplit model and GFS input data with 0.5° resolution with the corresponding source term shown in the top right panel of Fig. 3. The forward model run was set up in the same manner as the backward runs. The output concentrations presented in Fig. 6 are mean values in the layer between the surface and 100 meters above ground level. The results for the most likely location selected by the LS-APC-GS method are analogous.

The computed concentrations of I-131 are displayed in Fig. 6 for selected days, which are September 14 (left), October 14 (middle), and November 14 (right). The first two maps illustrate challenges for inverse modeling, since the aerosol was trans­ported to areas where no measurement data are available which corresponds well with reported measurements from Budapest and Praha in Fig. 1 where no measured activity is reported in Praha for the first half of the studied period. This also implies that the results may be very sensitive to the measurements from the station Budapest (denoted by the letter A in Fig. 6), which is the only station influenced in this case. This sensitivity will be studied in the next section.

The cumulated gamma dose for the whole 3-month period is computed for the most probable source terms computed using LS-APC-VB and LS-APC-GS methods. The cumulated gamma dose for the LS-APC-VB estimate is displayed in Fig. 7, left, for the Hysplit-GFS-0.5 model with the same settings as in the case of concentrations and for the LS-APC-GS estimate is displayed in Fig. 7, right, for the Hysplit-GFS-1.0 model. Results show that gamma dose amounts were largest in Hungary and Slovakia, while in the rest of Europe they were about two orders of magnitude smaller. However, Fig. 7 also shows that most of Europe was affected to some extent by the release. Notably, the simulation also shows that both the concentrations and dose amounts were very low even close to the release site. The maximum dose from the I-131 release during the studied 3-month period is approximately 0.001 mSv which is negligible, e.g., in comparison with the Czech natural radiation background of 3 mSv per year.

## 5.5    Sensitivity study

Since the distance between the measuring site in Budapest (denoted by the letter A in Fig. 2) and the IoI is only approximately 10 kilometers, the measured values at this station are often one order of magnitude higher than those from the other stations. Determining the source location and source strength could be thus dominated by the measurements from Budapest. However, simulating the concentrations at such a short distance is inaccurate since the meteorological input data are much coarser than the distance from the source to the station, and also the SRS calculations are done on a coarser grid. Thus, the errors of the source-location sensitivity can be relatively large, which may influence the estimated source term.

To test the sensitivity of the results to the values from the Budapest station, we run the source location excluding those measurements. The results are given in Fig. 8. The data in this case are much less informative, hence the uncertainty in source

location is much higher. Nevertheless, the maximum is mostly reached relatively close to the IoI facility. The maxima for individual dispersion models are 17.75° E 47.25° N for Flexpart-GFS-0.5, 18.75° E 48.25° N for Hysplit-GFS-0.5, 19.25° E 46.75° N for Hysplit-GFS-1.0, and 16.25° E 48.25° N for Hysplit-ECMWF-0.5 while the exact location of the IoI is approximately 18.96° E 47.49° N. Thus, even in this poorly informative case, the location is identified with very good accuracy. In all four cases, the uncertainty increased significantly to the south of the IoI where no measured data are available.

Source term estimates done without using Budapest data for the most likely locations for each dispersion model are given in Fig. 9. Full posterior densities are reported via their mean value (denoted by blue lines) and 95% highest posterior density regions (gray filled region). The source terms are accompanied by the computed total sum of activity with 95% uncertainty bounds. Overall, the total activities of estimated Flexpart-GFS-0.5, Hysplit-GFS-1.0, and Hysplit-ECMWF-0.5 source terms are on the same level as in the previous case where measurements from Budapest are included while the Hysplit-GFS-0.5 result is reduced approximately six-times; however, note that the maximum of the log-likelihood is no longer reached by Hysplit-GFS-0.5 but by Hysplit-GFS-1.0 where the total activity of the source term is estimated as 526 GBq with uncertainty bounds [444, 608] GBq which is in the same order of magnitude as the reported amount 342 GBq. Notice in particular that the reported peak related to the period October 12–14 is well captured by the LS-APC-VB algorithm with Hysplit-GFS-1.0 while this is not the case for the other models. Moreover, the release time profiles are different, with some peaks missing due to very low responses to these releases at the distant sensors especially in the first half of the studied period. This can be understood when considering concentrations in Fig. 6 and measurements from Budapest and Praha in Fig. 1. It can be seen that on the example day 14 September, the whole released activity is transported south-east of the release site where no measurement stations are available except the Budapest station which is not used in this sensitivity study. Notice in particular that station Praha did not measure any activity during this period (Fig. 1). This was the case also on many other days and explains why the LS-APC-VB algorithm does not produce any releases in September and the first half of October in all Flexpart and Hysplit model runs when the Budapest station is excluded.

Similar results are obtained using the GS method, Fig. 10. The estimated profiles correspond well with those obtained by the VB method, however, the associated uncertainty bounds are more realistic.

## 6   Conclusions

Low concentrations of iodine I-131 were detected in the atmosphere over Central Europe in the fall of 2011. After investigation, it was reported that I-131 was released from the Institute of Isotopes Ltd, Budapest, Hungary. In this study, the measurements of I-131 concentrations from several countries in Central Europe from fall 2011 were analyzed using two state-of-the-art dispersion models, Flexpart and Hysplit driven with three different meteorological input data sets (four model configurations in total), and latest Bayesian techniques of source term estimation and source location. We used these techniques to retrieve both the source location as well as the magnitude and temporal variation of the release, assuming that neither the release location nor the source strength was known. The results correspond well with the true location of the source where all four estimates are within one grid cell from the true location. The retrieved total emissions of I-131 have large error bounds and

also deviate between the different models and methods of source term estimation (Variational Bayes versus Gibbs sampling). The most likely estimate of the source term was 636 GBq with 90% confidence interval $[365, 1434]$ GBq. The reported total released dose 342 GBq is near the first percentile of the most likely posterior distribution. The time variation of the estimated source term is also in agreement with all aspects of the official report. Forward model simulations using the retrieved source

term showed that large areas of Europe were affected by the release but air concentrations and total dosages of I-131 were well below regulatory limits everywhere and the situation did not pose a health risk.

The performance of the Bayesian methodology was also tested when using less informative data. For this, we removed the most informative measurements from the nearest measurement station. Even in this case, the algorithm was able to locate the source with high accuracy but with significantly higher uncertainty, and the source strength was particularly uncertain. The

10 main reason for this large uncertainty was that all available measurement data (except for those taken at the one close-by station) were collected to the north of the release location. Therefore, releases could not be detected by this network during periods with northerly winds. This demonstrates the importance of the spatial distribution of measurement stations.

*Acknowledgements.* This research is supported by EEA/Norwegian Financial Mechanism under project MSMT-28477/2014 Source-Term Determination of Radionuclide Releases by Inverse Atmospheric Dispersion Modelling (STRADI). The authors would like to thank all the

15 laboratories who provided monitoring data.

## Appendix A: Truncated Gaussian distribution

Truncated normal distribution, denoted as $t\mathcal{N}$, of a scalar variable $x$ on interval $[a; b]$ is defined as

$$t\mathcal{N}_x(\mu, \sigma, [a, b]) = \frac{\sqrt{2}\exp((x - \mu)^2)}{\sqrt{\pi}\sigma(erf(\beta) - erf(\alpha))}\chi_{[a,b]}(x), \tag{A1}$$

where $\alpha = \frac{a - \mu}{\sqrt{2}\sigma}$, $\beta = \frac{b - \mu}{\sqrt{2}\sigma}$, function $\chi_{[a,b]}(x)$ is a characteristic function of interval $[a, b]$ defined as $\chi_{[a,b]}(x) = 1$ if $x \in [a, b]$

and $\chi_{[a,b]}(x) = 0$ otherwise. $\mathrm{erf}()$ is the error function defined as $\mathrm{erf}(t) = \frac{2}{\sqrt{\pi}}\int_0^t e^{-u^2}\,\mathrm{d}u$.

The moments of truncated normal distribution are

$$\langle x \rangle = \mu - \sqrt{\sigma}\frac{\sqrt{2}[\exp(-\beta^2) - \exp(-\alpha^2)]}{\sqrt{\pi}(\mathrm{erf}(\beta) - \mathrm{erf}(\alpha))}, \tag{A2}$$

$$\langle x^2 \rangle = \sigma + \mu\widehat{x} - \sqrt{\sigma}\frac{\sqrt{2}[b\exp(-\beta^2) - a\exp(-\alpha^2)]}{\sqrt{\pi}(\mathrm{erf}(\beta) - \mathrm{erf}(\alpha))}. \tag{A3}$$

For multivariate case, see (Tichý and Šmídl, 2016).

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

| Measuring site | Geographic coordinates | Number of measurements | Laboratory |
|---|---|---|---|
| Budapest | 47°25'N, 19°20'E | 12 | NRIRR |
| Alt-Prerau | 48°48'N, 16°28'E | 1 | AGES |
| Retz | 48°45'N, 15°57'E | 1 | AGES |
| Usti nad Labem | 50°40'N, 14°02'E | 13 | SUJB |
| Ostrava | 49°50'N, 18°17'E | 12 | SURO |
| Ceske Budejovice | 48°58'N, 14°28'E | 14 | SUJB |
| Praha | 50°04'N, 14°27'E | 16 | SURO |
| Gdynia | 54°31'N, 18°32'E | 12 | CLRP |
| Sanok | 49°33'N, 22°12'E | 12 | CLRP |
| Katowice | 50°16'N, 19°01'E | 12 | CLRP |
| Zielona Gora | 51°56'N, 15°31'E | 12 | CLRP |

**Table 1.** List of the sampling sites from which I-131 measurements were used in this study. NRIRR - *National Research Institute for Radiobiology and Radiohygiene* (regular on-site radiological measurements in NRIRR, http://www.osski.hu/info/ks/ksv_en.html)*, Hungary*; AGES - *Austrian Agency for Health and Food Safety, Austria*; SUJB - *State Office for Nuclear Safety* (data retrieved from the Monitoring of Radiation situation database, MonRaS, http://www.sujb.cz/monras/aplikace/monras_en.html)*, Czech Republic*; SURO - *National Radiation Protection Institute, Czech Republic*; CLRP - *Central Laboratory for Radiological Protection, Poland.*

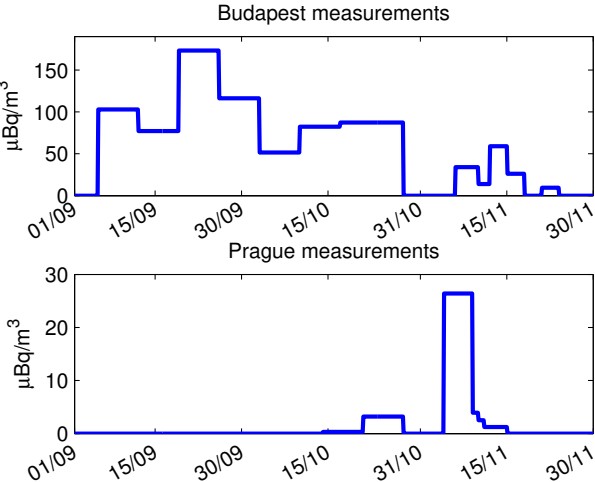

**Figure 1.** Measurements of I-131 activity concentrations in ambient air made at the stations Budapest (top) and Praha (bottom) displayed via their daily mean concentration.

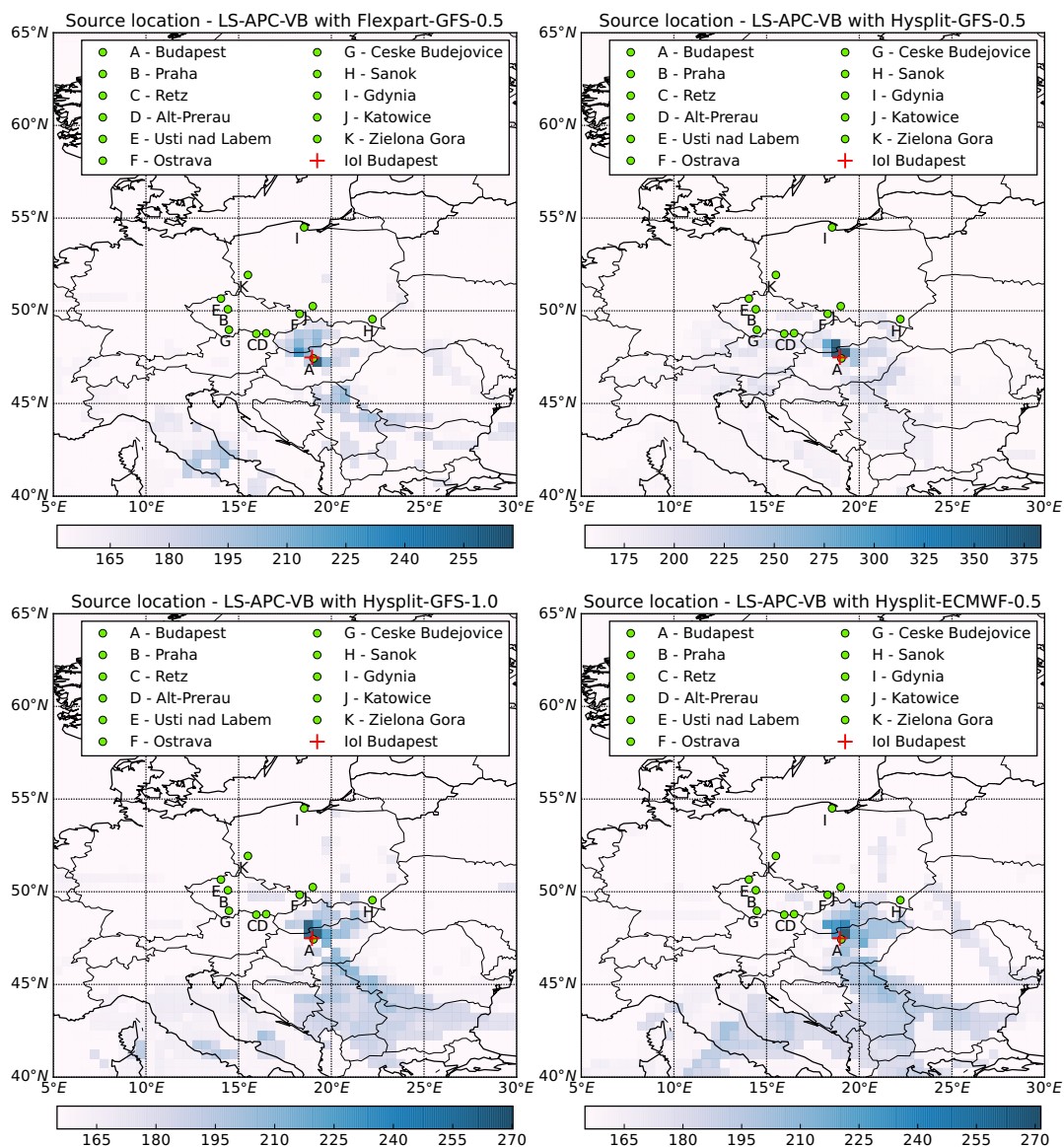

**Figure 2.** Source location via marginal log-likelihood where the observed data are explained by a release from a grid cell using the LS-APC-VB algorithm for all four tested combinations of dispersion model and meteorological data: Flexpart-GFS-0.5 (top left), Hysplit-GFS-0.5 (top right), Hysplit-GFS-1.0 (bottom left), and Hysplit-ECMWF-0.5 (bottom right). The measuring sites (a list is given in Table 1) are displayed using green circles while the location of the Institute of Isotopes (IoI) Ltd is displayed using a red cross.

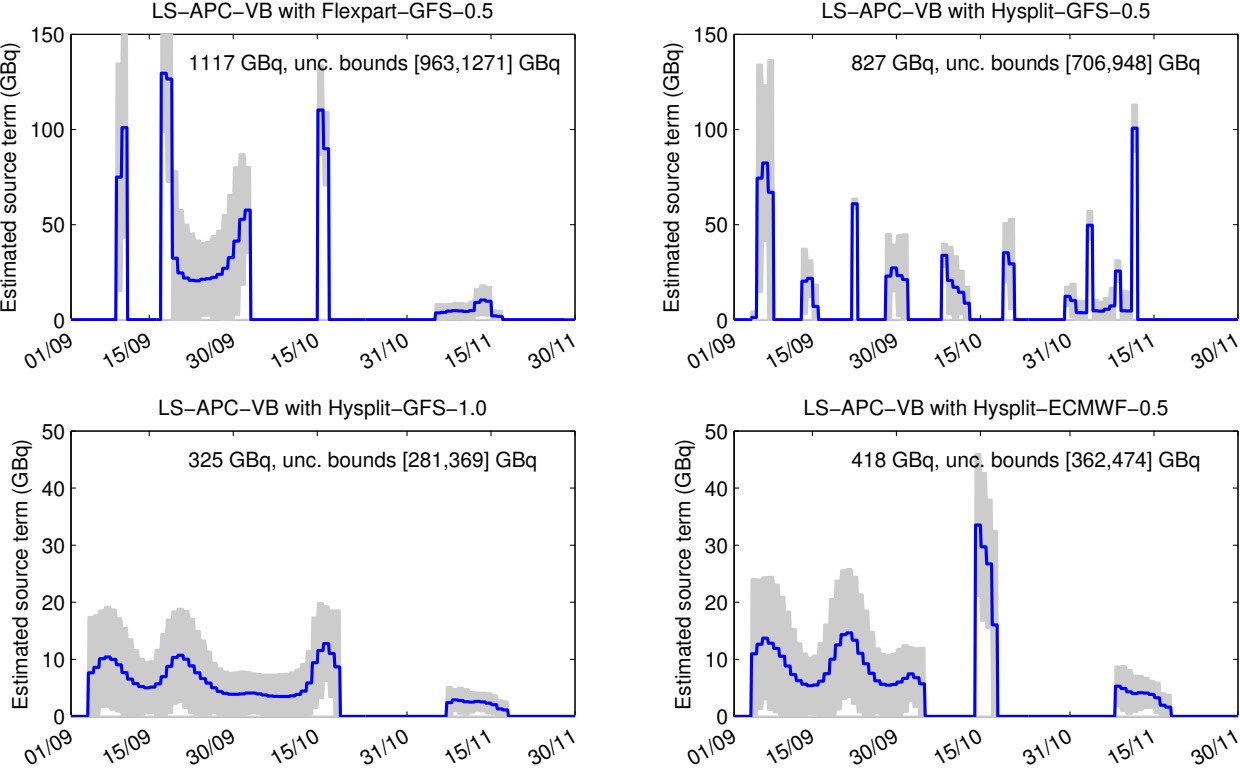

**Figure 3.** Estimated source terms at locations selected by the marginal likelihood method, shown in Fig. 2, using the LS-APC-VB algorithm for all four tested combinations of dispersion models and meteorological data : Flexpart-GFS-0.5 (top left), Hysplit-GFS-0.5 (top right), Hysplit-GFS-1.0 (bottom left), and Hysplit-ECMWF-0.5 (bottom right). The estimated source terms are accompanied by the 95% uncertainty regions (gray filled regions). The estimated activity for the whole period is reported inside each plot.

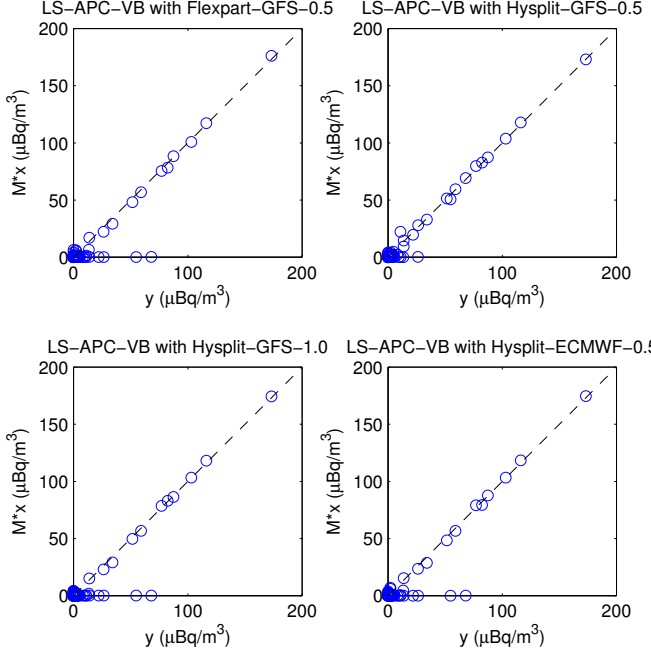

**Figure 4.** Scatter plots of the measurements **y** and the reconstructed signal $M\mathbf{x}$ using the LS-APC-VB algorithm with Flexpart-GFS-0.5 (top left), Hysplit-GFS-0.5 (top right), Hysplit-GFS-1.0 (bottom left), and Hysplit-ECMWF-0.5 (bottom right). The reconstructions are given for the estimated source locations, shown in Fig. 2, and the mean values of the estimated source terms, shown with blue lines in Fig. 3.

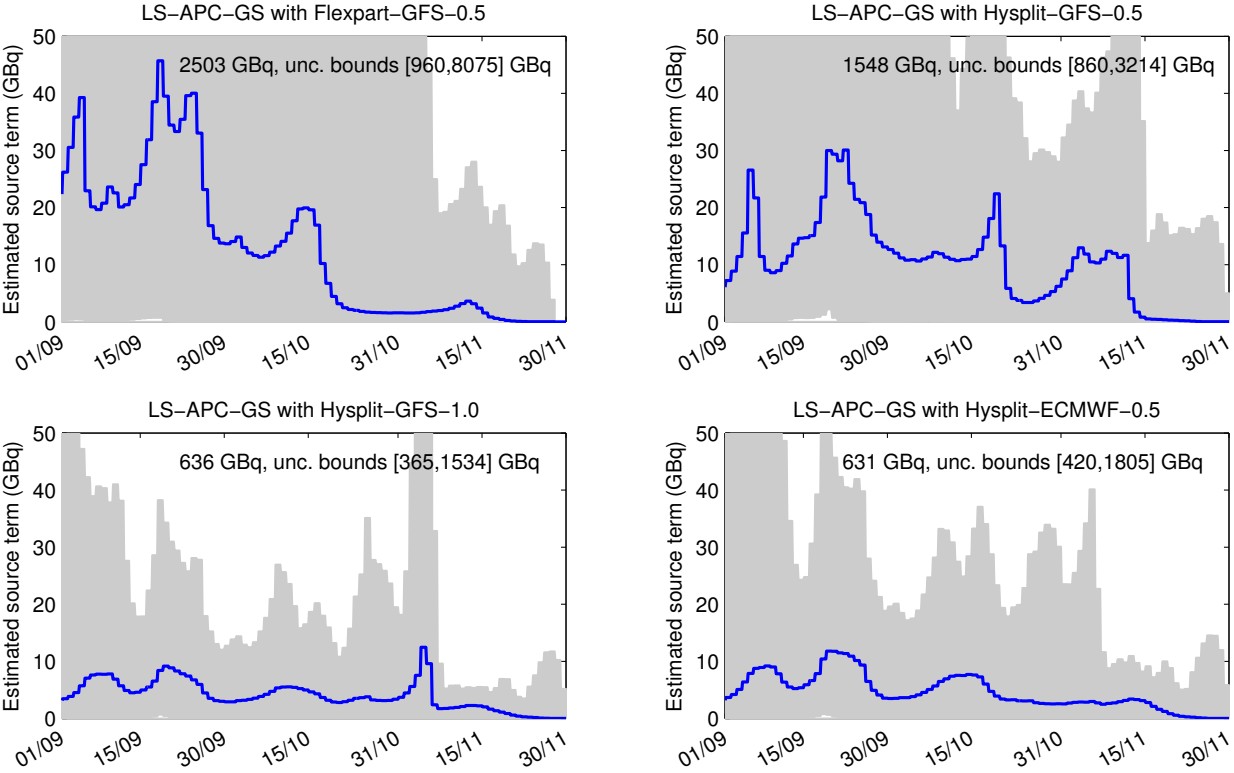

**Figure 5.** Estimated source terms at locations selected by the marginal likelihood method, shown in Fig. 2, using the LS-APC-GS algorithm for all four tested combinations of dispersion models and meteorological data: Flexpart-GFS-0.5 (top left), Hysplit-GFS-0.5 (top right), Hysplit-GFS-1.0 (bottom left), and Hysplit-ECMWF-0.5 (bottom right). The estimated source terms are accompanied by the 95% uncertainty regions (gray filled regions). The estimated activity for the whole period is reported inside each plot.

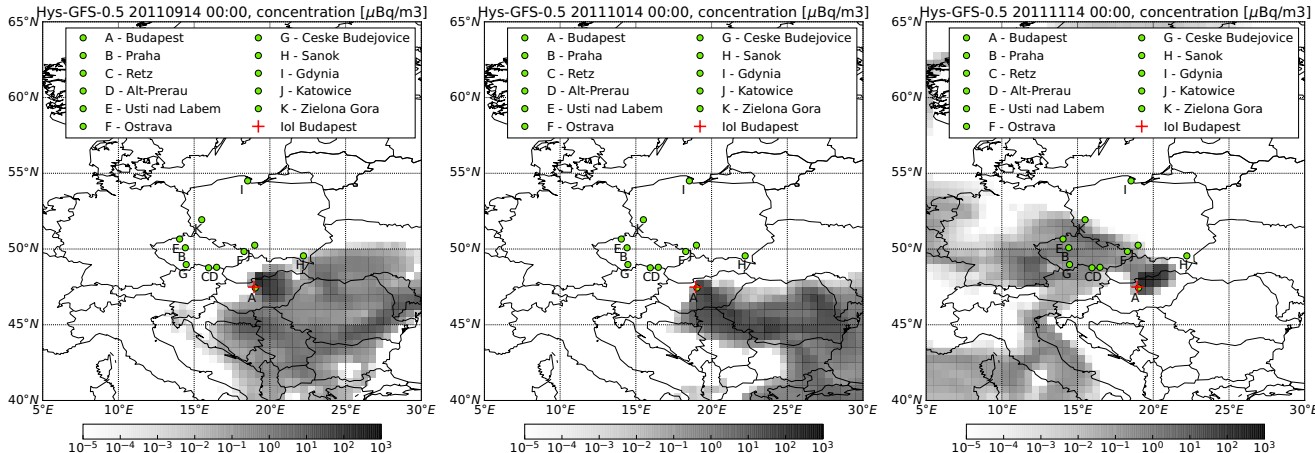

**Figure 6.** Maps of daily concentrations of I-131 for selected days (dates are reported at the top of each panel, September 14 in the left panel, October 14 in the middle, and November 14 in the right panel) using the Hysplit model with GFS input data with 0.5° resolution and with the source term computed using the LS-APC-VB algorithm given in Fig. 3, top right.

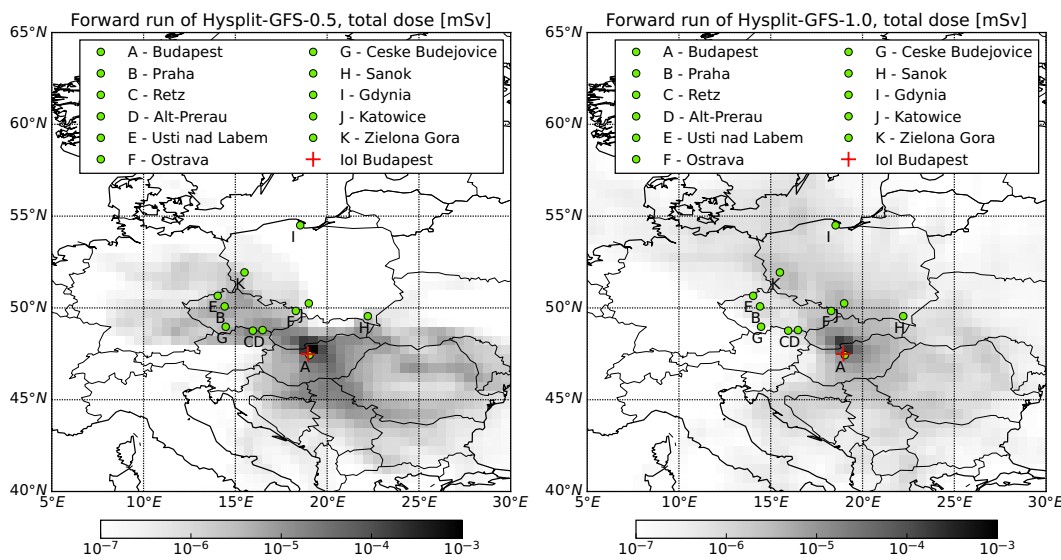

**Figure 7.** I-131 total dose for the whole 3-month studied interval. Left: simulation using the Hysplit model with GFS input data with 0.5° resolution and with the source term computed using the LS-APC-VB algorithm given in Fig. 3, top right. Right: simulation using the Hysplit model with GFS input data with 1.0° resolution and with the source term computed using the LS-APC-GS algorithm given in Fig. 5, bottom left.

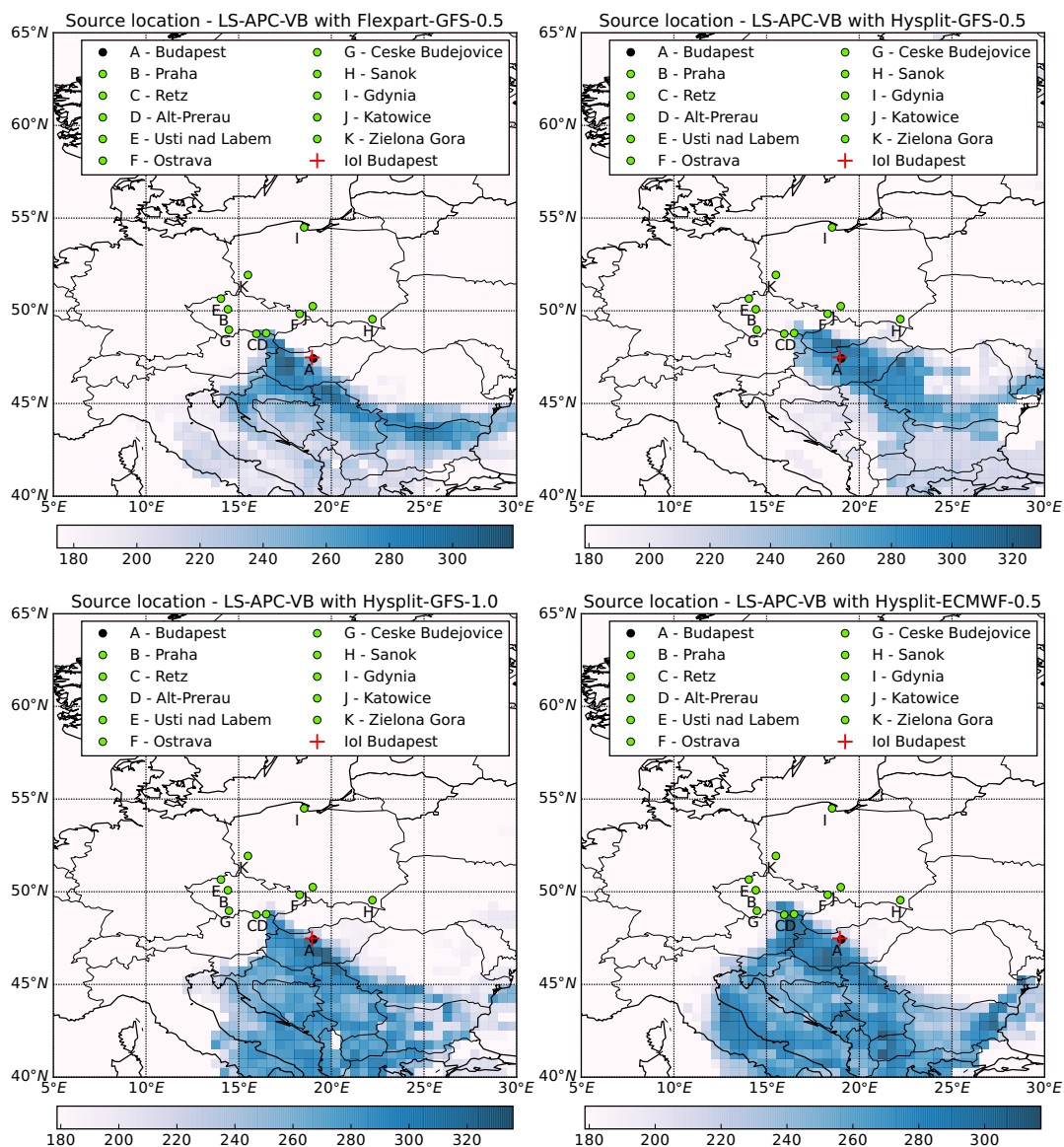

**Figure 8.** Sensitivity study of the source location using measurements without the Budapest station. Marginal log-likelihood that the observed data are explained by a release from a grid cell using the LS-APC-VB algorithm for each tested combination of dispersion model and meteorological data: Flexpart-GFS-0.5 (top left), Hysplit-GFS-0.5 (top right), Hysplit-GFS-1.0 (bottom left), and Hysplit-ECMWF-0.5 (bottom right). The measuring sites (see Table 1) are displayed using green circles while the location of the Institute of Isotopes (IoI) Ltd. is displayed using a red cross and the excluded measuring station Budapest (denoted by A) is displayed using a black circle.

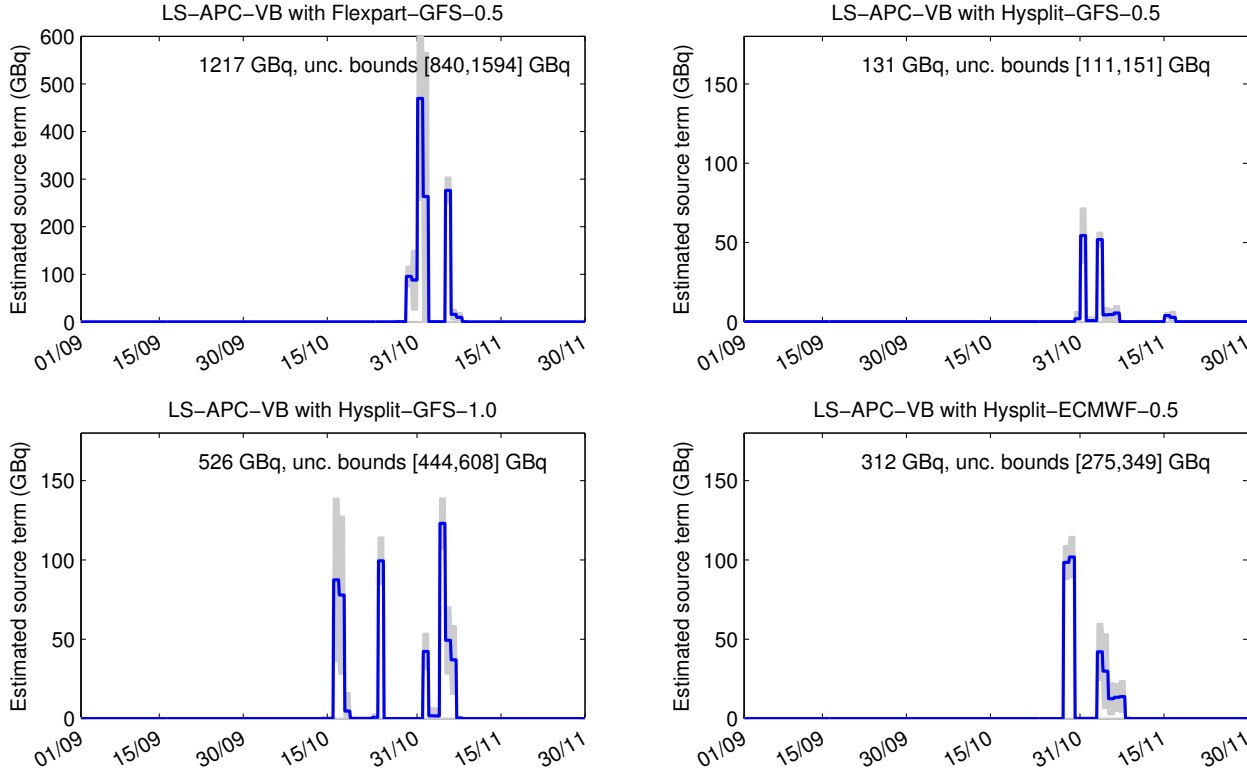

**Figure 9.** Estimated source terms at locations selected by the marginal likelihood method, shown in Fig. 8, with excluded measurements from Budapest using the LS-APC-VB algorithm for all four tested combinations of dispersion models and meteorological data: Flexpart-GFS-0.5 (top left), Hysplit-GFS-0.5 (top right), Hysplit-GFS-1.0 (bottom left), and Hysplit-ECMWF-0.5 (bottom right). The estimated source terms are accompanied by the 95% uncertainty regions (gray filled regions). The estimated activity for the whole period is reported inside each plot.

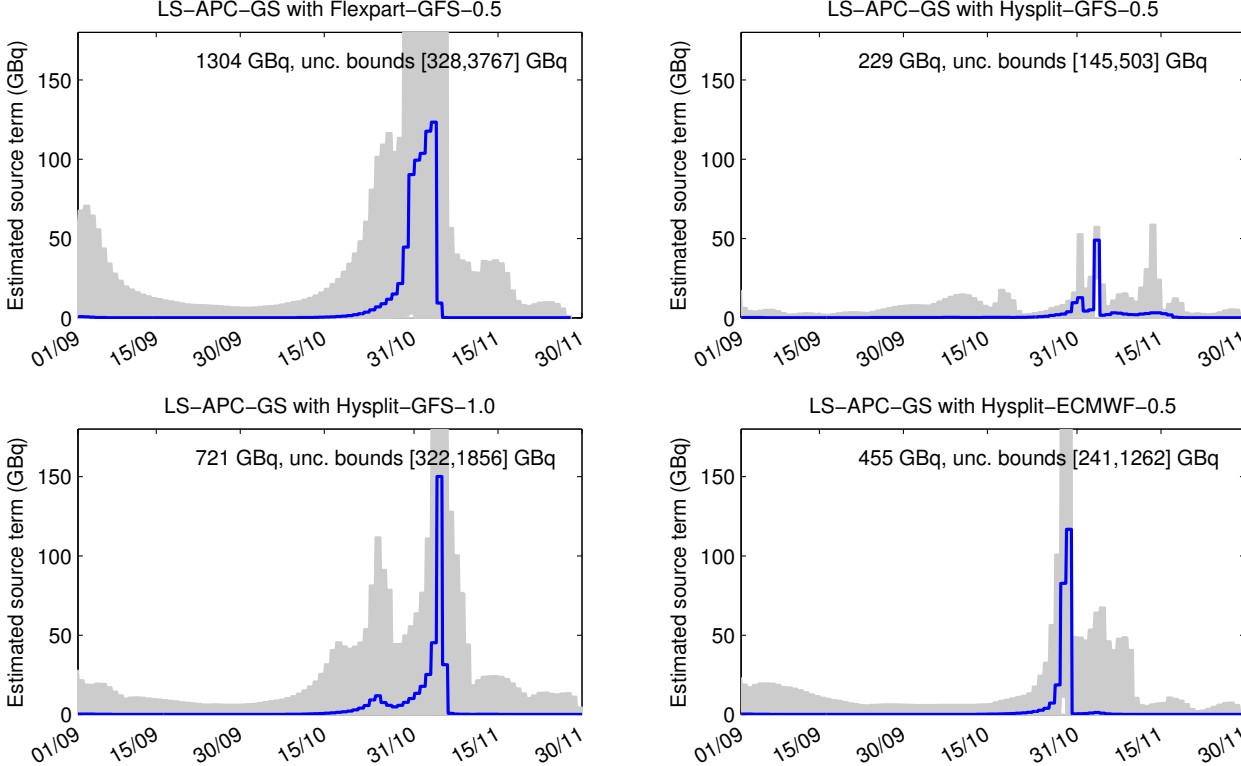

**Figure 10.** Estimated source terms at locations selected by the marginal likelihood method, shown in Fig. 8, with excluded measurements from Budapest using the LS-APC-GS algorithm for all four tested combinations of dispersion models and meteorological data: Flexpart-GFS-0.5 (top left), Hysplit-GFS-0.5 (top right), Hysplit-GFS-1.0 (bottom left), and Hysplit-ECMWF-0.5 (bottom right). The estimated source terms are accompanied by the uncertainty regions of the 5th and 95th percentile (gray filled regions). The estimated activity for the whole period is reported inside each plot.