# Peer review of "Bayesian inverse modeling and source location of an unintended I-131 release in Europe in the fall of 2011"

_Atmospheric Chemistry and Physics, 2017_

## Referee Comment (RC1) · W. Raskob (Referee) · 11 Apr 2017

As pointed out in my initial review, there is one issue not fully discussed in the paper. The total amount released was well reconstructed. The time dependency not at all – assuming that most was released during only two days. The models, however, proposed significant releases much earlier. The question that should be discussed further is the reason for this. Is it an artefact? Is the meteorological data together with the limitation of these models the reason for this? The timing of the source term is even more important of the total amount released as this determines the areas affected and the countermeasures needed.

the new approach can only be evaluated if such a discussion is performed, otherwise

the results might be just achieved by chance.

---

## Author Comment (AC1) · 22 Jun 2017

We would like to thank you for providing us with detailed reviews of our paper. We have considered all the comments and notes and we are glad that we can submit a revised version of our paper. In the following text, we will respond to all comments.

**Tichy et al. present an analysis of the I-131 release from the Institute of Isotopes in Hungary during the Fall of 2011. They use a sparse set of atmospheric observations and attempt to infer the source location and magnitude. The paper is reasonably well written but the figures could use quite a bit of improvement. In terms of content, the authors have currently not evaluated the meteorology**

[Figure]

used for this work which strikes this reviewer as a major short-coming. I would recommend major revisions for this manuscript.

**1  Specific Comments:**

**1.  Evaluation of the meteorology: The largest concern I have with this manuscript is the lack of meteorological evaluation. The authors use two different particle dispersion models (FLEXPART & HySplit) but both are driven by the same meteorology (GFS at 0.5 ◦ × 0.5 ◦ resolution). Obviously, if the meteorology is incorrect then the authors will obtain erroneous results. Surely there are observations of windspeed and direction the authors could use to evaluate. Further, the authors assume that all of the I-131 was release in particulate form and, as such, will undergo both dry and wet deposition. How well does GFS simulate the clouds and precipitation? It seems that this would be crucial to accurately representing the dry and wet deposition. Particle dispersion models are strongly dependent on the accurate simulation of the mixed layer heights because this typically dictates the sensitivity of the particles to surface fluxes, how well does GFS simulate mixed layer heights? A systematic bias in mixed layer heights could easily lead to SRS matrices that are too sensitive (or too weak), this would then manifest itself in a bias in the source magnitude.**

- Sensitivity of to the meteorological conditions is indeed essential. We have now added comparison with another source of data, specifically ECMWF at 0.5°×0.5°resolution. All data sets we use are reanalysis data, i.e. their assimilation with all available observations has already been done by the data provider. Since, we use continental weather fields for the period of three months, an attempt to validate this dataset is beyond the scope of this paper. We aim to use a finite set of existing data, and compare similarities and differences of the inversion

results for each of them.

- It is true that a systematic error in meteorological conditions may also imply bias of the results. In our experiments, the differences in results for different weather fields are however smaller than between different dispersion models.

Changes made in the paper:

1. We have computed the inversion model on a different set of meteorolologial data from ECMWF and compared the results with those obtained for the GFS data.

2. We have extended the discussion on possible influence of the meteorological data on the systematic overestimation of the released dose.

**2. Choice of Inversion Method?: As I mentioned in my pre-review, the authors have intelligently framed the problem by examining just a single source and re-peating the analysis for an ensemble of potential sources. However, it's unclear to this reviewer why they need to: (a) use Bayesian Model Selection (see Minor Comment #1) and (b) they they need to use LS-APC. Their system of equations is very small (117 × 91) and could be solved on a laptop, so why don't they use a more flexible framework like a hierarchical Bayesian with an MCMC, rjMCMC, or CMA-ES? Additionally, do their results provide an error estimate? It seems like many of their reported numbers are missing error bars which makes it difficult to evaluate. A prime example of this is in the abstract when comparing the reported emission (342 GBq) and their derived emission (490 GBq), if the error bars are large then these numbers may be statistically indistinguishable.**

- The reason for the choice of LS-APC is that it is a hierarchical Bayesian method that estimates its own nuisance parameters (hyper-parameters) from the data.

In effect, the method estimates different set of hyper-parametes for each location. The Bayesian models selection is a theoretically justified measure how to compare results with different priors. We show its effect in response to Minor Comment #1.

- An oversimplification may be the use of Variational Bayes method to evaluate the posterior of the LS-APC model. The reason is the need for fast evaluation of all 10000 inversion runs (4 models times 2500 map tiles). However, the LS-APC model can be also solved using Monte Carlo, specifically the Gibbs sampler (GS) which provides more accurate representations of the posterior distribution. Since it is more computationally demanding, we used it only on much smaller number of locations. Since the results are global approximation of the posterior, we consider these estimates of the release profile to be the main outcome of the paper.

- Both methods VB and GS provide uncertainty bounds, but the VB methods is known to underestimate them. We now report the uncertainty bounds in the abstract.

Changes made in the paper

1. Description of the Gibbs sampling was added as Section 3.2.2. and the method was applied to a reduced number of locations. Estimates of the release profiles at the selected locations are now compared to those from the VB approximation. The GS results combine more potential solutions of the problem yielding larger uncertainty bounds. Due to larger tails of the posterior distribution the median of the total release dose is higher that that from the VB method.

2. Reporting of the results in the whole paper has been extended to report the uncertainty bounds.

**2 Minor Comments**

**1. Bayesian Model Selection: Section 3.2 is confusing and it's not clear what the reader is supposed to gain from this section. They introduce a new variable (M) then do not use it. They introduce an equation (Eq. 10) with 2 terms and then simplify it to just exp(LMi ). The authors then introduce a complicated set of equation (Eq. 11) but do not explain the variables or terms (they point the reader to the supplement where they, again, do not explain the variables or terms). It's unclear to this reviewer why the authors need to use this variational lower bound when they should have posterior probabilities (that was supposed to be the motivation for using this LS-APC). Their system of equations is very small (117 × 91), why can't the authors just solve for the source magnitude at each potential location and compare the probabilities of those solutions? It seems that that would greatly simplify the problem.**

- We agree that description of the model selection in the main paper and the supplement was hard to follow. We have rewritten both parts.

- The comparison of magnitudes (total mass, used e.g. by Bocquet (2007)) may be misleading since it is highly dependent on the regularization parameter. In our approach, the regularization terms potentially differ for each location making them mutually incomparable. The marginal likelihood is a principled way how to compare them.

- The marginal likelihood is indeed a complex result with many terms. Explanation of the effect of each term is not very clear since it is based on lower bound of an integral measure. The terms mutually balance influences of the data and the prior terms. Sensitivity study using simulations is beyond the scope of this paper. Simplification of the full marginal likelihood for model comparison is possible, however, usually at the loss of accuracy. For comparison, we now provide maps

of the marginal likelihood (left) and its first term which is the norm of the model residues (right) in Fig. 1.

Note that the map of residues also provides a good source location, however, with some additional artifacts that are not present on the map provided by the full marginal likelihood.

Changes made in the paper

1. The section on model selection and the supplementary material were rewritten to improve clarity of presentation.

**2. Figures: None of the figures are particularly well done. Most labels are small and difficult to read. The spatial maps are particularly difficult to read. Red and green are very hard to pick out on a dark gray background. Most of the panels in the line plots and scatter plots could be combined into single panels, this would greatly facilitate comparison between the sensitivity studies.**

- We have changed the colormap of the maps to have monotonic appearance. Also, we have increased the font size of the labels in the Figures.

- Due to the extended number of compared models, combination of the maps and scatter plots in a single figure is now problematic. However, we keep the same layout of the studied models in all Figures.

**3. Discussion of Previous Work: The authors have omitted to mention a range of other studies that have used made major strides to address similar problems (e.g., objectively determining hyper-parameters and using particle dispersion models to estimate sources). At the bare minimum, most of the citations in the**

**inverse modeling section should include "e.g.,". There are, quite literally, text-books written about most of this work. An example of this is their Wotowa et al. citation in the first paragraph of Section 3. This is a review paper that cites hundreds of other papers for that particular application of particle dispersion models to estimate sources (which is already a small subset of the literature on this work). Omitting the "e.g.," is misleading.**

- Indeed, we are citing only subset of literature related to parameter estimation and atmospheric modeling and the use of e.g. is more appropriate.

Changes made in the text

- We added more references to SRS matrix description (Seibert, 2001; Wotawa et al., 2003; Seibert and Frank, 2004) and I also added "e.g." to this and several to Introduction. Moreover, we added reference to the latest review paper by Hutchinson et al. (2017) .

**3 Specific comments**

**1. Page 4, Line ∼30: How long is the sampling time? It seems that this could be very problematic.**

- The sampling time varies between 3-7 days which is regular sampling time of the stations. Indeed this limits accuracy of the results. Therefore, estimation of daily releases from these data is not well conditioned. The uncertainty is now well represented by the results of the Gibbs sampling method. We also added measurements from Budapest and from Prague for illustration.

**References**

M. Bocquet. High-resolution reconstruction of a tracer dispersion event: application to ETEX. *Quarterly Journal of the Royal Meteorological Society*, 133(625):1013–1026, 2007.

M. Hutchinson, H. Oh, and W.-H. Chen. A review of source term estimation methods for atmospheric dispersion events using static or mobile sensors. *Information Fusion*, 36:130–148, 2017.

P. Seibert. Iverse modelling with a lagrangian particle disperion model: application to point releases over limited time intervals. In *Air Pollution Modeling and its Application XIV*, pages 381–389. Springer, 2001.

P. Seibert and A. Frank. Source-receptor matrix calculation with a Lagrangian particle dispersion model in backward mode. *Atmospheric Chemistry and Physics*, 4(1):51–63, 2004.

G. Wotawa, L.-E. De Geer, P. Denier, M. Kalinowski, H. Toivonen, R. D'Amours, F. Desiato, J.-P. Issartel, M. Langer, P. Seibert, et al. Atmospheric transport modelling in support of ctbt verification—overview and basic concepts. *Atmospheric Environment*, 37(18):2529–2537, 2003.

[Figure]

[Figure]

**Fig. 1.** Source location using LS-APC-VB model selection (left) and using model residues (right).

---

## Author Comment (AC2) · 22 Jun 2017

We would like to thank you for providing us with detailed reviews of our paper. We have considered all the comments and notes and we are glad that we can submit a revised version of our paper. In the following text, we will respond to all comments.

[Figure]

**1  Specific Comments:**

**As pointed out in my initial review, there is one issue not fully discussed in the paper. The total amount released was well reconstructed. The time dependency not at all – assuming that most was released during only two days. The models, however, proposed significant releases much earlier. The question that should be discussed further is the reason for this. Is it an artefact? Is the meteorological data together with the limitation of these models the reason for this? The timing of the source term is even more important of the total amount released as this determines the areas affected and the countermeasures needed.**

**The new approach can only be evaluated if such a discussion is performed, otherwise the results might be just achieved by chance**

- It is not clear from the official report how were the released amounts obtained and what are their uncertainty bounds. Unfortunately, on the reported days of the main release, the released activity was monitored only by the Budapest station (see Figure 6 middle in the new manuscript). We have now added a Figure with measured concentrations in Budapest and Prague in the analyzed period. Note that the concentrations in Budapest indicate that the release in September was not negligible and there is not evidence for abnormal release in early October.

- The temporal profile reconstructed by the Variational Bayes may be misleading since it finds only local approximation of the posterior distribution. We have added another approximation of the posterior distribution based on Gibbs sampling. This approximation evaluates global approximation of the posterior and thus it provides more realistic representation of uncertainty in the timing. The conclusion is that the available data can provide only rather wide bounds on the release profile.

Changes made in the paper

1. A figure with measured concentrations from Budapest and Prague has been added. Predicted concentrations on October 14 are now reported in Fig. 6.

2. Reconstructions of the temporal profiles were also done by the Gibbs sampling which provides more accurate approximation of the true posterior.

3. Discussion on the influence of the weather conditions on the results has been added.

**2 Initial Review**

**The paper discusses one of the most important issues in emergency management and response, the source term. As the source term in a progressing accident is typically not known, source term reconstruction capabilities as essential for a good response. Even long time later, the source term might be not fully known and retrospective analysis is important. This paper address an event from 2011 where iodine was released into the atmosphere. The approach presented is well described and a promising method to estimate the source term based modelling and monitoring records.**

**The paper is well structured and describes the results very well. There are some issues to be considered in a possible revision of the paper.**

**Comment 1: Figure 1 is hard to read with the dark background.**

- Colormap of the maps has been changed.

**Comment 2: Figure 3 is presented in the backward modelling section but it refers to the forward modelling. Please clarify**

- Thanks for suggestion, this typo is now corrected.

**Comment 3: In the section of dose, the maximum dose is presented. I would not recommend to publish this as the cess size with about 45x55 km is too large to get the maximum dose captured which may happen close to the source point**

- In the figure, we present maximum average dose. Certainly the maximum dose can be different than the average. However, since the dose is summed over the period of three months, the difference between the average and the maximum should be minimal even on such a large cell.

**Comment 4: There is one issue not fully discussed in the paper. The total amount released was well reconstructed. The time dependency not at all – assuming that most was released during only two days. The models, however, proposed significant releases much earlier. The question that should be discussed further is the reason for this. Is it an artefact? Is the meteorological data together with the limitation of these models the reason for this? The timing of the source term is even more important of the total amount released as this determines the areas affected and the countermeasures needed.**

- This has been addressed by the use of Monte Carlo evaluation via Gibbs sampling. See discussion of the Specific Comment 1.

---

## Author Comment (AC3) · 22 Jun 2017

During preparation of the second revision, we have discovered an error in code for association of meassurements with the results of dispersion models. Only a few meassurements from one station were affected. This issue is now corrected in the revised paper.